# An Investigation of Customer Psychological Perceptions of Green Consciousness in a Green Hotel Context: Applying a Extended Theory of Planned Behavior

**DOI:** 10.3390/ijerph19116795

**Published:** 2022-06-02

**Authors:** Taeuk Kim, Jungwoo Ha

**Affiliations:** 1Department of Hotel & Restaurant Management, Kyonggi University, Seoul 03746, Korea; teokim1305@naver.com; 2Department of Tourism Event Management, Kyonggi University, Seoul 03746, Korea

**Keywords:** image perception of green corporate social responsibility, green psychological benefit, green consciousness, green-related attitude toward behavior, green-related subjective norm, green-related perceived behavioral control, green behavior, willingness to sacrifice for the environment

## Abstract

We investigated the relationship between green consciousness and green behavior, and the relationship between psychological state, attitude, and behavior of green hotel customers by applying variables suitable for an expanded theory of planned behavior. The purpose of the study was to predict green behavior based on the theory of planned behavior. Together with preceding research including the correlation between customers’ image perception of green corporate social responsibility (CSR), green psychological benefit, and green consciousness, we added willingness to sacrifice for the environment to define the relationship with green consciousness and green behavior. A survey was conducted with 410 customers of green hotels in Seoul, Korea more than twice over a period of over 6~12 months. Vague and insincere answers were removed. SPSS 18.0 and Amos 20.0 were used to conduct factor and SEM data analysis. Our theory was verified and adopted following validation from our analysis. The results have important theoretical and practical implications for the environment by providing primary data on customers’ perceptions of eco friendliness to support the establishment of corporate management strategies. Moreover, they may encourage green hotels to participate in preventing environmental problems.

## 1. Introduction

Two years after the COVID pandemic began in January 2020, a need for corporations to build sustainable growth systems to fulfill their environmental responsibility has arisen. Consequently, corporate social contributions to research and design (R & D) have emerged as a result [1].

The most remarkable adoption of this change occurred with hotels in the tourist industry. As “untact” services (customer encounters without face-to-face contact with employees) became the hot issue throughout the service industry, it became inevitable that strategic discussions exploring the issue would emerge to identify a solution for management, sales, marketing, public relations (PR), human resources (HR), meetings, incentives, conferences and exhibitions (MICE), tours, online travel agencies (OTA), and food and beverage services (F & B) [2,3]. The COVID-19 pandemic limited hotel marketing targets to domestic tourists [2] by collapsing the tour industry’s supply and demand. This change required a more detailed marketing approach by business holders, with keywords being premium, luxury, privacy, eco-friendly, wellness and moreover [4]. In other words, the pandemic pushed hotel businesses to focus on customers’ interests and affections instead of retaining a hold on the old property-centered goods and marketing approach [5]. This led to the following conclusion; a “unique quality” of the hotel that had previously attracted customers’ interest and feeling became the current core of the past hotel model that focused on luxury. It is now time to bring in hygiene PR to relieve customers’ need for quarantine measures and relief. Thus, it is clear that corporations should capture the generational needs and apply them to products with haste through continuous environmental protection, donation, and win-win buildup with the regional economy [6].

This can signify the purpose of a corporate eco-friendly image: that customers will take a positive view of the corporation [7,8]. For example, green hotels can use either the BCA “Green Mark” or “Leadership in Energy and Environmental Design” (LEED) certification to project an eco-friendly image of its venture to customers to enhance customer satisfaction and eco-friendly behavior. [9]. Customers’ perceptions, attitudes, and behavior regarding the corporation can be affected by its image [10], and it has been shown that an eco-friendly, corporate social responsibility (CSR) hotel image can also affect the eco-friendly perceptions and behavior of its customers [11]. In other words, greater awareness of the eco-friendly CSR hotel can encourage customers to show greater green concern and activity as a result of their green hotel experience [12]. Moreover, a green hotel experience can provide guests with green psychological benefits (GPB) in both direct and indirect ways [13]. According to [14,15], green CSR image and its benefits can have a positive effect on peoples’ perceptions and behavior. Thus, we can say that corporate image and benefits generate an awakening of the customers’ concerns from a primordial level.

In this study, we needed to examine hotel customers’ eco-friendly behavior from various points of view (i.e., eco-friendly image, eco-friendly GPB, green concern), similar to those already verified [16,17,18,19], Meng et al., 2016. Here, we differentiated verification by identifying customers’ eco-friendly behavior with three components: willingness to pay premium price (WPPP) [20]; green hotel revisit intention [21]; and green consuming intention [22] to offer more detailed result variables that could define the correlation of explanatory variables.

We also added one of the most accurate theories of anticipatory eco-friendly attitude and behavior [23,24]—the theory of planned behavior (TPB)—to confirm customers’ eco-friendly behavior. Moreover, we added new variables to the TPB [25,26,27,28] to verify the extent of influence. In this study, the new variable is the willingness to sacrifice for the environment (WSE) [29,30,31], which will expand the TPB to predict WPPP, intention to visit an environmentally responsible hotel (IVER), and sustainable consumption intention (SCI). It was confirmed that WSE has the potential to increase green behavior (GB), WPPP, IVER, and SCI.

In summary, our first aim was to examine the relationship between image perception of green corporate social responsibility (IP-GCSR) and GPB. Next, we defined the structural relationship between GB and the expanded TPB with consciousness in terms of green-related attitude toward behavior (GATB), green-related subjective norms (GSN), green-related perceived behavioral control (GPBC), and WSE.

## 2. Theoretical Background

### 2.1. Green Hotel and Image Perception of Green Corporate Social Responsibility

The hotel industry is undertaking a huge role in environmental pollution. According to the United States Environmental Protection Agency (EPA), the hotel industry is in the top five industries based on CO2 emissions. Kasavana [32] defined an environmental hotel (hereafter referred to as green hotels; GHs) as “a contribution to economy by saving water and energy, reducing waste. Savior of the earth and people, dedicated by sensible/active managers,” while Al Suwaidi et al. [33] defined GHs as “an active organization focuses on standard hotel profits plus social responsibility, performing environmental protection and behavior”. According to Chan and Hsu [34], it is an attempt to minimize the hotel industries’ negative effect on the environment. Thus, the focus of these hotels is as follows: waste/towel/linen recycling, low-pressure shower and washbowl facilities, water-free urinals, replaceable bath supplies, automatic room climate control, light sensors, and natural ventilation. However, this could force guests to face compromised luxury, convenience, and quality as a sacrifice for eco protection [35], and could place hotels in an unfavorable position in terms of their competition [36]. Thus, GHs need to set two values on scale: one for environmental protection and one for sacrifice of service quality and luxury.

The effect of the eco-friendly products’ social benefit can be described as the fulfillment of corporate social responsibility (CSR) or meeting the corporate expectations of the entire society. With regard to corporate citizenship, customers expect corporations to be positive influencers in society by fulfilling their social duties [37]. When corporations do meet their expectations through GB, such as eco-production for a positive influence, this increases customers’ GPB as their role expectation is satisfied [38].

Corporate socially responsible activities create an identity for the organization that offers a continuous and unique value to the individuals within. Following this logic, we can say that IP-GCSR can have the same effect on the organization [39]. It can also be said that a value congruence will occur when corporate socially responsible activities occur, as the company and individuals within it will share the value; consequently, individuals’ evaluations will be positive [40]. In the context of eco-production, using eco-friendly products will lead customers to equate their benefit with the value of the eco-friendly product; this will be shown as a positive response to eco-friendly production. When the benefits of purchase/consumption of eco-friendly products are considered as private benefits, a casual relationship between them will satisfy the social exchange theory principle of reciprocity [41,42]. Thus, customers will make more positive evaluations of certain products if their concern for eco-friendly products in terms of psychological benefit (i.e., satisfaction from a sense of altruism), economic benefit (i.e., saving cost and energy), and functional benefit (i.e., skin improvement, non-toxicity to the body) increase through the principle of reciprocity [43,44].

### 2.2. Image Perception of Green Corporate Social Responsibility, Green Consciousness, and Green Psychological Benefit

Mohammed & Al-Swidi [45] define GHs as “Hotel [s] that perform eco-friendly policy or green management with social responsibility to protect environment but keeping the existing hotel value or service quality”. The purpose of GHs is to preserve cultural and natural resources by providing environmental responsibility to both customers and employees and making a contribution to the local community and activating it. This will lead to improving the local community’s wealth while simultaneously securing corporate profits [46]. Environmental protection and green management can draw a positive impression from customers who are aware of environmental issues. This can work as a crucial factor to activate hotel customers’ purchase action [47].

In this context, we can state that positive IP-GCSR will increase customers’ purchase intention and have a favorable effect in acquiring their trust [48]. In terms of the relationship between green management, IP-GCSR, and competitiveness, it is shown that corporations under eco-friendly management haver higher competitiveness compared with a control group, and an even better corporate image (which can affect customers’ perceptions of service quality). It can also promote customers’ purchase activity and increase their competitive power [49]. Thus, we can say that IP-GCSR can affect customers’ evaluation of the corporation. Basically, customers’ GPB comprises three psychological benefits: warm glow, self-expressive benefits, and natural experience. We focused on natural experience in this study as it is the most suitable and adoptable factor for our target group: hotel customers’ green concern.

People generally wish to spend time with nature to restore their physical and psychological health [50]. Experiencing nature can offer and cultivate emotional happiness and positive feelings by relieving stress and aggression [51]. This is why eco-friendly brands try to show beautiful natural scenes when promoting their products with promotions that speak of comfort and harmony. This is important because it can build the basis of a natural experience [52]. In 2008, Hartmann and Apaolaza-Ibáñez [43] mentioned that green concern can induce GB. Natural experiences can be the core factor in building such concern and behavior, which builds an eco-friendly value/attitude and perception. This will have a positive effect on GB [53].

Bashir et al. [54] defined the meaningful relationship between GPB (both mental and functional), IP-GCSR, and eco-friendly loyalty, by surveying Malaysian hotel customers. Other previous research [41,55,56,57,58,59] has verified the positive effect of customers’ perceived GPB on their value, attitude, and perception. From this preceding logic, we developed our hypotheses as follows:

**Hypothesis** **1** **(H1).**
*The image perception of GCSR has a significant effect on GC.*


**Hypothesis** **2** **(H2).**
*GPB has a significant effect on GC.*


### 2.3. Green Consciousness, Green-Related Theory of Planned Behavior, and Green Behavior

To understand how people adopt GCs, we need to place a great deal of effort into increasing peoples’ WSE by achieving environmental sustainability [60]. According to Baldassare and Katz [61], people who are aware of air or water pollution are more likely to reduce private vehicle utilization and increase their GATB, such as recycling, water saving, or purchasing green products. GC can be explained as the key factor leading to GATB. Thus, the greater the concern about damage to nature, the greater their GATB to relieve the environmental risk. It is also verified that GC can affect GATB [57,62,63]. It is therefore crucial to examine GC from various different perspectives. Accordingly, we consider the definition of environmental risk in order to increase GATB.

It is not easy to encourage people to take eco-friendly actions rather than just being concerned about it. This can be explained by a mismatch between comprehension and action about public activity. If public action provides zero or low benefit, GATB may be compromised [64]. The relationship between GC and GATB can be explained by social dilemma theory, which describes a gap between social benefit and individual advantage [65]. For example, if the environment is considered private property that is owned by everyone, people will take extra care and spend more to maintain its condition. However, most people do not consider the environment to be privately owned; therefore it is hard to see such investment taking place. This kind of mismatch and dilemma is the main threshold between GCs and GB.

Many theories predict the relationship between GCs and GB. The TPB is one of the most efficient theories to explain the connection and has been verified by various studies [13,27,66,67,68,69].

The first core component of the TPB is attitude towards behavior. Attitude is a learned tendency against a certain target or activity, regardless of whether it is positive or negative. It works as a fine measurement unit for evaluation of certain behaviors by displaying extreme evaluation or emotion against it [70]. The second component is the subjective norm. This can be defined as the social affect or pressure from a person’s important relation, such as a family member or colleague, to force certain action; in other words, “support or resist from one’s relatives against one’s action” [71]. The third component is perceived behavioral control. This is willpower to control one’s own will or the perception of certain activities’ difficulty. According to Ajzen [23], it refers to one’s level of belief about one’s own sense of control. Behavior intention can be categorized into two types: unfavorable or favorable. More inclusively, this can be considered as purchase intent/word of mouth intent/complaint behavior intent and price sensitivity. When someone purchases a product or service, experience of use can shape attitude toward the product and control future activity. This activity can trigger the will to revisit, review, or recommend to others in the future [72].

Currently, the TPB is an extension model due to the limitations of the existing theory of reasoned behavior [23]. To make the TPB more reliable, more variables were added [23] and its superiority has been acknowledged, as it offers better prediction of behavior intention than the past TPB [73]. Existing research [28,74,75,76] has shown the capability of extended TPB, verifying its use with the new variables to make better predictions. In the field of tourism research, value-belief-norm theory (VBNT) was applied to add green knowledge [77], perceived authenticity and environmental concern [78], and perceived moral obligation [16] to perform better examination.

Specifically, preceding research with TPB-applied verification for environmental consciousness and behavior are used to probe tourists’ revisit intention for eco-friendly destinations [66], hotel customers’ GB [79], and WPPP for GHs derived from GC [80]. Following this logic, we developed our hypotheses as follows:

**Hypothesis** **3** **(H3).**
*GC has a significant effect on GB.*


**Hypothesis** **4** **(H4).**
*GC has a significant effect on GTPB.*


**Hypothesis** **4a** **(H4a).**
*GC has a significant effect on GATB.*


**Hypothesis** **4b** **(H4b).**
*GC has a significant effect on GSN.*


**Hypothesis** **4c** **(H4c).**
*GC has a significant effect on GPCB.*


**Hypothesis** **5** **(H5).**
*Green-related TPB has a significant effect on GB.*


**Hypothesis** **5a** **(H5a).**
*Green-related ATB has a significant effect on GB.*


**Hypothesis** **5b** **(H5b).**
*Green-related SN has a significant effect on GB.*


**Hypothesis** **5c** **(H5c).**
*Green-related PBC has a significant effect on GB.*


### 2.4. Green Consciousness, Willingness to Sacrifice for the Environment, and Green Behavior

Where a person willingly renounces life, property, renown, or profit for someone or for a certain purpose, this is referred to as willingness to sacrifice [81]. When someone surrenders something of their own or dedicates themself to a certain situation, this is called sacrifice. Renouncing happiness or participating in the dedication of life and property are regarded as such a situation [82]. Willingness to sacrifice refers to both monetary and non-monetary sacrifice [83]. While it may be true that renouncing can incur great personal material loss, it is also true that others’ evaluation can incur much higher valuation than the real cost. Such choices can only be found in public service or altruistic acts (which are hard to perform), as it can bring such rare self-accomplishment that other individuals cannot even imagine it [84]. As the concept of sacrifice has broadened to domains such as sociology, political science, and business administration, research collaboration across each domain has provided alternative explanations. For example, in sociology, sacrifice is explained as an investment of one’s most valuable thing to join a group or community [85]. In business administration, it has been seen as a voluntarily absorption of organizations’ structural error or imperfection [86].

Research has considered the concept of WSE [87,88,89], showing that tourists’ intention to visit GHs is the most important variable in WSE. Other research has been undertaken to predict GB. Surveying museum visitors to check their WSE has shown that their will to sacrifice has great value in activating personal norms and increasing GATB [85]. Schultz et al. [90] have suggested that customers who support green products are likely to have relatively high environmental or ecological interest. Furthermore, a study targeting college students by Davis et al. [29] verified the mediator effect of eco-friendly devotion on the relationship between WSE and self-satisfaction. This means that customers with higher eco-friendly devotion are more likely to have WSE, causing a meaningful effect on WSE. According to Oreg and Katz-Gerro [91], participants from 26 countries demonstrated a mediating effect of WSE on the relationship between environmental concern and eco-friendly citizenship. WSE also demonstrates a mediating effect on the relationship between biospheric value and GATB in [19], surveying EH visitors. Lastly, Han et al. [31] verified the relationship between WSE and GB (i.e., reducing waste, recycling) with incompatible results for males (negative effect) and females (positive effect). In this study, we examine the relationship between GC, WSE, and GATB with preceding research, according to the following hypotheses:

**Hypothesis** **6** **(H6).**
*GC has a significant effect on WSE.*


**Hypothesis** **7** **(H7).**
*WSE has a significant effect on GB.*


## 3. Method

### 3.1. Measures and Questionnaire Development

In this study, we applied an online questionnaire presented in two sections. The first part of the survey requested demographic information (i.e., gender, age, marital status, level of education, job career). In the second part, the conceptual model was constructed using IP-GCSR, GPB, GC, GTPB, GATB, GSN, GPBC, GB, WSE, WPPP, IVER, and SCI. To assess the study constructs, the measures were employed from the existing literature [17,19,92,93,94]. Multi-items with a five-point scale were utilized to evaluate the constructs.

In particular, three items were adopted from [45] to assess IP-GCSR, eight items were adopted from [92] to assess GC, and 10 items were adopted from [25] to assess GATB (4), GSN (3), and GPBC (3) of the TPB. In addition, four items used to assess GPB were adopted from [16], five items used to assess WSE were adopted from [18], three items used to assess IVER were adopted from [21], three items used to assess WPPP were adopted from [20], and three items used to assess SCI were adopted from [22].

The questionnaire was improved through a pre-test with graduate students and faculty members majoring in hospitality and tourism. It was then confirmed and reviewed by academics and industry experts. All measurement items used in the present research are shown in Appendix A.

Before conducting the research, procedural remedies were used to reduce common method bias (CMB) as follows: The accuracy of the questionnaire’s content and adequacy of its items were checked by 10 hotel management PhDs between 2 March 2022 to 9 March 2022. Moreover, the questionnaire was designed to protect respondent anonymity and reduce evaluation apprehension [25,95].

The survey was conducted between 10 March 2022 and 9 April 2022, targeting customers who made at least one-to-two visits to hotels in a six-month period in Seoul, South Korea. Online (mobile) and offline (by researcher) surveys were administered simultaneously, resulting in 410 samples for analysis after classifying the 500 samples distributed.

### 3.2. Data Collection Process and Demographic Profiles

To secure the effectiveness of the study, we surveyed GH customers to evaluate correlations between GC, GPB, IP–GCSR, GTPB, and GB, using non-probability convenience sampling.

Before commencing with the survey, we offered a concrete explanation (i.e., either a Green Mark or LEED certificated hotel) and a picture of EHs to help the subjects’ understanding of IP–GCSR. Responses were met by a 5-point Likert scale to determine sincerity and honesty. Drink vouchers were offered as an incentive for participation.

### 3.3. Data Analysis and the Sample

To analyze the data, we used SPSS 20 and AMOS 20. As recommended by [96], the measurement model with a confirmatory factor analysis (CFA) was initially evaluated before assessing the proposed structural model. Reliability and construct validity of measurement items for each construct were evaluated. structural equation modeling (SEM) was then conducted. A chi-square difference test was employed for modeling comparison. Lastly, a test for metric invariance was utilized in order to evaluate the hypotheses.

Regarding the demographic characteristics of the 410 participants, 53.2% (n = 410) were female customers and 46.8% (n = 410) were male customers. Moreover, approximately 55.7% (n = 410) of the respondents were single, while 55.61% were unmarried (n = 228) and 44.39% were married (n = 182). In addition, 32.68% (n = 134) of respondents were 30–40 years old, 26.34% (n = 108) of respondents were 40–49 years old, 23.90% (n = 98) of respondents were 20–29 years old, 7.56% (n = 31) of respondents were 50–59 years old, and 9.51% (n = 39) of respondents were 60 years old and above. Their levels of education were high school graduate and below (13.90%, n = 57), two years of college (28.78%, n = 118), four years of college (40.98%, n = 168), and postgraduate or higher (16.34%, n = 67).

## 4. Results

### 4.1. Common Method Bias

This study intended to mitigate possible common method variance (CMV) by using two procedural remedies in the survey design and using the analysis method. First of all, this study used different cover stories when the respondents answered using the same scale to facilitate the psychological separation between criterion variables and predictors. For example, the cover story for the pro-environmental concern scale was “The following statements are irrelevant to the above questions. Please read carefully each statement, then mark from extremely disagree to agree with your recent impressions”. Another method of reducing CMV was dealing with item ambiguity, as suggested by [97]. For example, this study provided specific definitions of imprecise terms to help respondents’ comprehension. Moreover, the results of Harman’s one-factor analysis, which is a post hoc test to detect possible CMV [97], showed that the CMV of the unmeasured latent methods factor was 1.1% and the one-factor measurement model was confirmed to fit the data satisfactorily (goodness-of-fit statistics for the measurement model: χ^2^ = 1002.679, df = 685, *p* < 0.001, χ^2^/df = 0.683, RMSEA = 0.033, CFI = 0.952, IFI = 0.952, TLI = 0.945).

This study made an effort to reduce CMV by employing procedural remedies in the survey design stage. Therefore, CMV did not influence the parameter estimations.

### 4.2. Confirmatory Factor Analysis

The confirmatory factor analysis (CFA) was performed to verify reliability and validity. As a result of the measurement model, Goodness-of-fit statistics for the measurement model χ^2^ = 160.222, df = 87, *p* < 0.001, χ^2^/df = 0.543, RMSEA = 0.045, CFI = 0.993, IFI = 0.983, TLI = 0.982, were judged to be excellent overall (Byrne, 2013). Factor loading, significance probability of *t*−value, average variance extracted (AVE), and construct reliability (CR) were checked to ensure the convergent validity of the latent variables of the measurement model. The confidence coefficients (Cronbach’s α) of factor loading were shown to be between 0.642 and 0.997, which was more significant than the 0.6 suggested by [98]. Moreover, AVE values and CR values ranged from 0.563 to 0.905 and from 0.806 to 0.966, respectively. These values were all greater than the level of 0.5 and 0.7 suggested by [96].

In addition, a correlation analysis was performed as shown in Table 1 to verify discriminant validity. As a result of Pearson’s correlation analysis, all variables of IP-GCSR, GPB, GC, GATB, GSN, GPBC, WSE, and GB were *p* < 0.05, indicating a significant correlation association [99]. Thus, discriminant validity was confirmed.

### 4.3. Structural Model and Hypothesis Testing

In this study, the GB (WPPP, IVER, SCI) of green hotel customers was investigated based on extended TPB theory. The structural equation model (SEM) analysis was generated by using the maximum likelihood estimation method as an estimation method for both model and procedures’ evaluation [100]. Goodness of fit of Structural Model (χ^2^ = 1241.952 df = 677, *p* < 0.001, χ^2^/df = 0.545, RMSEA = 0.037, CFI = 0.990, IFI = 0.980, TLI = 0.989) was satisfactorily higher than the standard value.

Moreover, SEM had shown high prediction power for R^2^(GC) = 0.121, R^2^(GATB) = 0.320, R^2^(GSN) = 0.198, R^2^(GPBC) = 0.200, R^2^(WSE) = 0.190, R^2^(GB) = 0.328 and *t*-values and standardized path coefficient were shown as the result in Table 2 and Figure 1. The path estimates show that IP-GCSR had a significantly positive effect on GC (*β* = 0.319, *t* = 6.338 ***), GPB had a significantly positive effect on GC (*β* = 0.110, *t* = 2.275 **). Thus, H1 and H2 were supported. Moreover, GC had a significantly positive effect on GB (*β* = 0.244, *t* = 4.416 ***). Thus, H3 was supported. GC had a significantly positive effect on GATB (*β* = 0.180, *t* = 3.214 ***), GSN (*β* = 0.314, *t* = 5.502 ***), GPBC (*β* = 0.140, *t* = 2.533 *). Thus, H4(a-c) was supported. On the sub-factor of TPB, GATB had a significantly positive effect on GB (*β* = 0.246, *t* = 4.315 ***), GSN had a significantly positive effect on GB (*β* = 0.439, *t* = 7.277 ***), GPBC had a significantly positive effect on GB (*β* = 0.156, *t* = 2.659 *). Thus, H5(a-c) were supported. In addition, GC has a significantly positive effect on WSE (*β* = 0.120, *t* = 1.986 *). Thus, H6 was supported. Finally, WSE had a significantly positive effect on GB (*β* = 0.188, *t* = 3.178 ***). Thus, H7(a–c) were supported.

## 5. Discussion and Implications

### 5.1. Conclussions

In this study, we confirmed the effect of IP-GCSR and GPB on GC and their correlation. Predictors suggested by the TPB (i.e., attitude, subjective norm, and perceived behavioral control) were also checked for their effect on GHs and GB. After conducting the analysis, WPPP, IVER, and SCI were confirmed as subfactors of GB. Moreover, we added WSE to the extended TPB as a variable to predict outcome variables. To do so, we surveyed people who visited GHs in South Korea and performed data analysis. It is believed that it would be meaningful to investigate the relationship between the influence variables related to the image and perception of consumers visiting eco-friendly hotels in Korea.

As a result of verifying the relationship between TPB and GB in this study, [16,101] applied TPB supported this study hypothesis as a significant positive influence relationship. In addition, WSE validated in the study of [19] supported a significant influence relationship with GB (WPPP, IVER, SCI), but in the study of [31], it was found that WSE as an independent variable had a negative (−) effect on waste reduction and recycling intent. Therefore, Rahman and Reynolds [19] was supported by the hypothesis of this study, where WSE was shown as a significant positive (+) influence on GB, but [31] showed contradictory results. Therefore, studies are being conducted to apply ETPB by adding various variables to TPB, and it is believed that by adding WSE in this study, ETPB can provide basic evidence data as a part of predicting GB.

### 5.2. Theoretical Implications

First, this study identified the influence of GB, WPPP, IVER and SCI on EH and elucidated customers’ complex behavior concerning GHs. It was shown that customers have a greater tendency to visit GHs [19,69,102], although they hesitate to pay the premium price that result from the hotels’ eco-friendly policies [62]. This study explained an increase in customers’ GB, WPPP, IVER, and SCI in relation to GHs. The process expanded the understanding of how customers’ WPPP could overcome the price premium paradox of GHs.

Secondly, this study applied the TPB to GB, WPPP, IVER, and SCI toward EHs and expanded the research field. Previous research had attempted to explain complex consumer behavior related to GHs consumer behavior [15,47,87,103]; and it is believed that there are academic implications in that the approach to predict that eco-friendly behavior has been further expanded by applying TPB’s theory in previous studies.

### 5.3. Managerial Implications

First, GHs expose their concept and activity to customers in the context of ecology and environmentally-friendly management. Moreover, they should request that customers accept the inconvenience or price premium derived from GHs eco-friendly activity and increase their IVER and SCI. Furthermore, GHs need to establish a marketing position of environmentally sustainable management to visitors. The process can produce a positive attitude among customers toward GHs [104]. Thus, “positive attitude” toward GHs can be described as WPPP, IVER, and antecedents of SCI. The exposure of GHs concept and activity to the public and the increase in customers’ positive attitude can provide the actual solution to overcome GHs price premium paradox.

Second, GHs should perform marketing strategies to increase the sub-factors of the extended TPB; according to previous research, antecedents of SCI, WPPP, and IVER are the sub-factors (attitude, SN, GPBC, and WSE) of the extended TPB. For instance, high GH prices can determine customers’ GPBC. If the price of GHs becomes too high, customers will produce low GPBC as a result. Thus, GHs are suggested to define the valid price line that customers can perceive.

Third, GH visitors should be informed of what GHs can contribute to the environment. The benefits of the GH can motivate its visitors to promote the GH to their community. This will increase both the GB and IVER among multiple subjects.

Fourth, efforts are required to increase public attention on GHs environmental effects. The green-related CSR image of the GH will facilitate customers’ positive activity toward GHs. Therefore, it is necessary to expose the effect of environmental protection on continuity and intensity. For example, the hotel industry needs to establish a positive CSR image by performing social responsibility activities and eco-friendly campaigns without overruling the government’s environmental policy. A GH positioning strategy and communication campaign program for environmental issues could increase concern for the environment and GHs. This can increase public attention and their perceived social norm level for GHs.

Fifth, the comparative analysis of the influence of the image and benefits of hotel companies on customer attitudes and behaviors will be very meaningful depending on Asian and Western cultures. For example, the CSR image of a hotel company can provide a direction for strategic vision setting in global chain hotel companies (Marriott, Hilton, Accor, etc.) located in European countries.

### 5.4. Study Limitations and Future Research

First, online surveys were performed with sufficient GH image material, whilst online questionnaires were not concerned with participant opinions.

Second, participants were people who visited GHs over a six-month period. A gap of this length could result in memory loss or failure to complete the survey. Concerning Section 1 and Section 2, the researcher might visit the GH and make a directional approach to the visitors. After the COVID-19 situation improved, a direct approach (i.e., an interview) could be conducted instead of an offline survey if the efficiency of the research is at stake.

Third, the purpose of the visit (i.e., business, trip, leisure) or the company (i.e., family, relation of lovers, friend, coworker) could affect the eco-friendly experience in terms of mental state, satisfaction, or attitude. Further research should consider such factors to classify the visitors in order to draw more accurate data.

Forth, the research was performed during a 14-day period during the pandemic. Awareness of COVID-19 could be a powerful variable; Thus, further research should be undertaken in a more timely manner.

## Figures and Tables

**Figure 1 ijerph-19-06795-f001:**
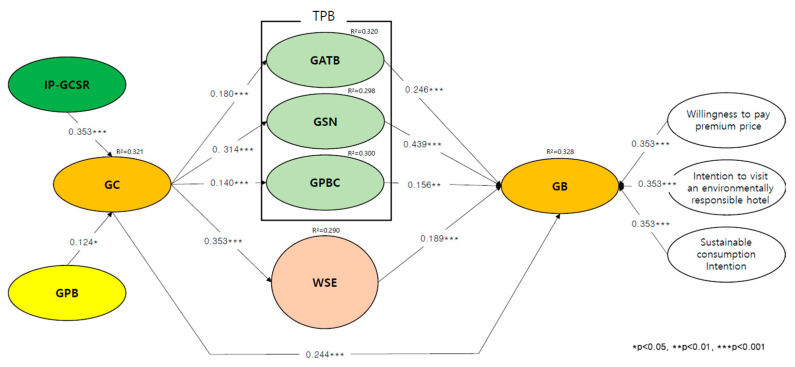
Structural equation model estimation and test for structural metric invariance. Note 1. Image Perception of Green Corporate Social Responsibility (IP-GCSR), Green Psychological Benefit (GPB), Green Consciousness (GC), Green related Theory of Planned Behavior (GTPB), Green related Attitude Toward Behavior (GATB), Green related Subjective Norm (GSN), Green related Perceived Behavioral Control (GPBC), Willingness to Sacrifice for the Environmental (WSE), Willingness to Pay Premium Price (WPPP), Intention to visit an environmentally responsible hotel (IVER), Sustainable Consumption Intention (SCI). Note 2. Goodness-of-fit statistics for the measurement model χ^2^ = 160.222, df = 87, *p* < 0.001, χ^2^/df = 0.543, RMSEA = 0.045, CFI = 0.993, IFI = 0.983, TLI = 0.982. Note 3. Goodness-of-fit statistics for the structural model: χ^2^ = 1241.952 df = 677, *p* < 0.001, χ^2^/df = 0.545, RMSEA = 0.037, CFI = 0.990, IFI = 0.980, TLI = 0.989.

**Table 1 ijerph-19-06795-t001:** The measurement model and correlation.

Construct and Scale Item		Standardized Loading	Mean (SD)	AVE (CR)	IP-GCSR	GPB	GC	GATB	GSN	GPBC	WSE	GB	√AVE
IP-GCSR	IP-GCSR1 IP-GCSR2 IP-GCSR3	0.746 0.881 0.809	4.11 (0.525)	0.662 (0.854)	1								0.814
GPB	GPB1 GPB2 GPB3 GPB4	0.785 0.871 0.723 0.642	4.07 (0.495)	0.577 (0.844)	0.633 ***	1							0.760
GC	GC1 GC2 GC3 GC4 GC5 GC6 GC7 GC8	0.695 0.756 0.772 0.766 0.748 0.723 0.742 0.795	3.90 (0.537)	0.563 (0.911)	0.075 *	0.042 *	1						0.750
GATB	GATB1 GATB2 GATB3 GATB4	0.727 0.847 0.854 0.652	3.93 (0.561)	0.600 (0.856)	0.032 *	0.058 *	0.665 ***	1					0.775
GSN	GSN1 GSN2 GSN3	0.678 0.788 0.815	3.99 (0.568)	0.581 (0.806)	0.021 *	0.014 *	0.581 ***	0.641 ***	1				0.763
GPBC	GPBC1 GPBC2 GPBC3	0.854 0.867 0.922	3.84 (0.689)	0.778 (0.913)	0.100 *	0.059 *	0.553 ***	0.430 ***	0.529 ***	1			0.882
WSE	WSE1 WSE2 WSE3 WSE4 WSE5	0.879 0.887 0.817 0.794 0.863	3.67 (0.714	0.720 (0.928)	0.158 ***	0.132 ***	0.542 ***	0.387 ***	0.438 ***	0.667 ***	1		0.849
GB	WPPP IVER SCI	0.893 0.997 0.878	3.60 (0.734)	0.905 (0.966)	0.072 *	0.124 *	0.367 ***	0.336 ***	0.358 ***	0.374 ***	0.364 ***	1	0.951

Note 1. SD = standardized deviation, AVE = average variance extracted, CR = composite reliability, image Perception of Green Corporate Social Responsibility (IP-GCSR), Green Psychological Benefit (GPB), Green Consciousness (GC), Green related Attitude Toward Behavior (GATB), Green related Subjective Norm (GSN), Green related Perceived Behavioral Control (GPBC), Willingness to Sacrifice for the Environmental (WSE), Green Behavior (GB); Note 2. Goodness-of-fit statistics for the measurement model: χ^2^ = 160.222, df = 87, *p* < 0.001, χ^2^/df = 0.543, RMSEA = 0.045, CFI = 0.993, IFI = 0.983, TLI = 0.982, * *p* < 0.05, *** *p* < 0.001; Note 3. All factors loadings are significant at *p* < 0.001.

**Table 2 ijerph-19-06795-t002:** Structural model results and hypothesis testing.

Hypothesized Paths	Coefficients		*t*-Values
H1: IP-GCSR → GC	0.353	<0.001	6.337 ***
H2: GPB → GC	0.124	0.024	2.258 *
H3: GC → GB	0.244	<0.001	4.416 ***
H4a: GC → GATB	0.180	0.001	3.214 ***
H4b: GC → GSN	0.314	<0.001	5.502 ***
H4c: GC → GPBC	0.140	0.011	2.553 **
H5a: GATB → GB	0.246	<0.001	4.315 ***
H5b: GSN → GB	0.439	<0.001	7.227 ***
H5c: GPBC → GB	0.156	0.008	2.65 9 **
H6: GC → WSE	0.094	0.074	1.987 *
H7: WSE → GB	0.189	0.001	3.178 ***
Explained variable:	R^2^(GC) = 0.121 R^2^(GATB) = 0.320 R^2^(GSN) = 0.198 R^2^(GPBC) = 0.200 R^2^(WSE) = 0.190 R^2^(GB) = 0.328		

Note 1. SD = standardized deviation, AVE = average variance extracted, CR = composite reliability, image Perception of Green Corporate Social Responsibility (IP-GCSR), Green Psychological Benefit (GPB), Green Consciousness (GC), Green related Attitude Toward Behavior (GATB), Green related Subjective Norm (GSN), Green related Perceived Behavioral Control (GPBC), Willingness to Sacrifice for the Environmental (WSE), Green Behavior (GB); Note 2. Goodness-of-fit statistics for the measurement model: χ^2^ = 160.222, df = 87, *p* < 0.001, χ^2^/df = 0.543, RMSEA = 0.045, CFI = 0.993, IFI = 0.983, TLI = 0.982, * *p* < 0.05, ** *p* < 0.01, *** *p* < 0.001; Note 3. All factor loadings are significant at *p* < 0.001.

## Data Availability

Data sharing is not applicable.

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
