# Peer review of "An Investigation of Customer Psychological Perceptions of Green Consciousness in a Green Hotel Context: Applying a Extended Theory of Planned Behavior"

_ijerph, 2022, doi:10.3390/ijerph19116795_

Round 1

Reviewer 1 Report

This manuscript addressed an important issue about first-hand research about the relationship between green consciousness and green behavior. This kind of research was vital to green hotel development under current special circumstance of “untact” service after the COVID pandemic began.  The study design was normative and the methods was appropriate. In the Introduction and Theoretical Background part listed more than thirty related studies, the authors listed 100 related studies. Compare the results with these peer studies in the Discussion part was quite important in the revised manuscript.

The necessary discussion I suggested be made in the '5.Results and Discussion' were:
(1) The roots and logic that the authors developed the 'Extended Theory of Planned Behavior' should be discussed first.
(2) Why the author choose the study areas "Seoul and Korea"? What characters of these two cities was suitable for Green Consciousness studies in a Green Hotel?
(3) If these results could be expanded for widely used in countries besides Asia, such as European countires? Or what extant should it be used elsewhere in Korean?

Author Response

Response to Reviewer 1 Comments

Manuscript ID:  ijerph-1737265
Title: An Investigation of Customer Psychological Perceptions of Green Consciousness in a Green Hotel Context: Applying a Extended Theory of Planned Behavior

Dear Editor and reviewers,

Thank you very much for reviewing our manuscript. We appreciate the opportunity you have afforded us to revise and resubmit. We found your suggestions to be thought-provoking and useful and have worked diligently to improve the manuscript as you suggested. Below, you will find our replies and responses to your constructive comments. Within the responses, red sections denote changes we made to the manuscript itself. Within the manuscript itself, changes are highlighted in red. We hope the changes made are satisfactory to you.

******************************************************************************************

Point 1: The roots and logic that the authors developed the 'Extended Theory of Planned Behavior' should be discussed first.

Response 1: Thank you for your comment. We have corrected the part pointed out by the reviewer correctly.

Point 2: Why the author choose the study areas "Seoul and Korea"? What characters of these two cities was suitable for Green Consciousness studies in a Green Hotel?

Response 2:  Thank you for your comment, and you raised good points. We have corrected the part pointed out by the reviewer correctly.

Point 3: If these results could be expanded for widely used in countries besides Asia, such as European countries? Or what extant should it be used elsewhere in Korean?

Response 3: Thank you for your comment, and you raised good points. We have corrected the part pointed out by the reviewer correctly.

Reviewer 2 Report

On page 7 you describe your sample that is non probabilistic, it looks as convenience. You should add some considerations about representativeness of this sample with reference to the target population.

 My main concern refers to your items that are on a five-point scale. In your analyses (FA, SEM), did you consider that these variables are ordinal? I have the impression that you did not since you calculated correlation coefficients.

Author Response

Response to Reviewer 3 Comments

Manuscript ID:  ijerph-1737265
Title: An Investigation of Customer Psychological Perceptions of Green Consciousness in a Green Hotel Context: Applying a Extended Theory of Planned Behavior

Dear Editor and reviewers,

Thank you very much for reviewing our manuscript. We appreciate the opportunity you have afforded us to revise and resubmit. We found your suggestions to be thought-provoking and useful and have worked diligently to improve the manuscript as you suggested. Below, you will find our replies and responses to your constructive comments. Within the responses, red sections denote changes we made to the manuscript itself. Within the manuscript itself, changes are highlighted in red. We hope the changes made are satisfactory to you.

******************************************************************************************

Point 1: On page 7 you describe your sample that is non probabilistic, it looks as convenience. You should add some considerations about representativeness of this sample with reference to the target population.

Response 1: Thank you for your comment. We have corrected the part pointed out by the reviewer correctly.

Point 2: My main concern refers to your items that are on a five-point scale. In your analyses (FA, SEM), did you consider that these variables are ordinal? I have the impression that you did not since you calculated correlation coefficients.

Response 2:  Thank you for your comment. We did these variables as ordinal.

Reviewer 3 Report

Overall, the study follows a rigid design to do the analysis step by step. Meanwhile, they adopt TPB and propose a framework to examine the green hotel initiative based on it. However, what lacks is a novel factor or dimension. Every factor here seems quite traditional and all measurements adopted are quite popular in relevant research fields. Here are some of my concerns:

1. The abstract is a bit personalized but not academic and it is too long as well. Keywords are too many.

2. Authors can provide the proposed framework in section 2 first to give the academia a panoramic view of the study first.

3. Part 3 is not suitable to use ‘materials’ and can be changed to “method’ alone.

4. Discussion part is too weak. Authors should open a dialogue between the finding of this study and previous studies to show whether there is any difference or anything new.

5. Authors only add one more dimension ‘WSE’ to TPB. Thus, is it appropriate to state that previous studies are limited in terms of factors considered while you just added one more factor to address the issue?

6. The managerial implication is a bit pushy and it is always not possible to request the customers to do something they dislike. The better tone should be ‘suggest’ or ‘advise’ them to do so.

Author Response

Thank you for your comment, our responses please see the attached file.

Round 2

Reviewer 1 Report

Following the previous suggestions, this manuscript has made progress to some extent, especially the Discussion and Implication section.

Author Response

thank you so much for your suggestions.

Reviewer 2 Report

ok for publication

Author Response

thank you so much for your suggestions.

Reviewer 3 Report

1. I hope the abstract can be condensed a bit and use simple words to highlight the meaning and the finding of the study.

2. Line 418 'the more specific results are as follows', for this statement, I do not quite get what the author means actually. 

3. After a search in google scholar, I find there are some studies utilizing TPB to investigate eco-friendly behavior in the area of tourism and hospitality. Thus, the implication of what mentioned in line 428-430 is not existent or at least not original. 

4. Is it correct to name section 5 'results and discussions? I can see results mainly in section 4 while section 5 is more like conclusion with discussions and implications. 

Author Response

Thank you so much for you suggections.
